# Chitooligosaccharide Conjugates Prepared Using Several Phenolic Compounds via Ascorbic Acid/H_2_O_2_ Free Radical Grafting: Characteristics, Antioxidant, Antidiabetic, and Antimicrobial Activities

**DOI:** 10.3390/foods11070920

**Published:** 2022-03-23

**Authors:** Ajay Mittal, Avtar Singh, Bin Zhang, Wonnop Visessanguan, Soottawat Benjakul

**Affiliations:** 1International Center of Excellence in Seafood Science and Innovation, Faculty of Agro-Industry, Prince of Songkla University, Hat Yai 90110, Thailand; ajy.mittal@yahoo.com (A.M.); avtar.s@psu.ac.th (A.S.); 2College of Food and Pharmacy, Zhejiang Ocean University, Zhoushan 316022, China; zhangbin_ouc@163.com; 3National Center for Genetic Engineering and Biotechnology (BIOTEC), National Science and Technology Development Agency, Khlong Luang 12120, Thailand; wonnop@biotec.or.th

**Keywords:** chitooligosaccharide, plant polyphenols, antioxidant, antimicrobial, antidiabetic

## Abstract

Chitooligosaccharide (COS)-polyphenol (PPN) conjugates prepared using different PPNs, including gallic, caffeic, and ferulic acids, epigallocatechin gallate, and catechin, at various concentrations were characterized via UV-visible, FTIR, and ^1^H-NMR spectra and tested for antioxidant, antidiabetic, and antimicrobial activities. Grafting of PPNs with COS was achieved. The highest conjugation efficiency was noticed for COS-catechin (COS-CAT), which was identified to have the highest total phenolic content (TPC) out of all the conjugates (*p* < 0.05). For antioxidant activities, DPPH and ABTS radical scavenging activities (DPPH-RSA and ABTS-RSA, respectively), oxygen radical absorbance capacity (ORAC), ferric reducing antioxidant power (FRAP), and metal chelating activity (MCA) of all the samples were positively correlated with the TPC incorporated. COS-CAT had higher DPPH-RSA, ABTS-RSA, ORAC, and FRAP than COS and all other COS-PPN conjugates (*p* < 0.05). In addition, COS-CAT also showed the highest antidiabetic activity of the conjugates, as determined by inhibitory activity toward α-amylase, α-glucosidase, and pancreatic lipase (*p* < 0.05). COS-CAT also had the highest antimicrobial activity against all tested Gram-negative and Gram-positive bacteria (*p* < 0.05). Overall, grafting of PPNs, especially CAT on COS, significantly enhanced bioactivities, including antioxidant and antimicrobial, which could be used to retard spoilage and enhance shelf-life of various food systems. Moreover, the ability of COS-CAT to inhibit digestive enzymes reflects its preventive effect on diabetes mellitus and its complications.

## 1. Introduction

Increased shrimp demand generates unavoidable solid waste during processing, especially cephalothorax and shell (50–60% of total weight). This waste shows deleterious impacts on the environment (i.e., pollution) if not properly managed. To attenuate the adverse effect of biowaste, shrimp shell has been valorized to various bioactive compounds such as protein hydrolysate, astaxanthin, and chitin (Nirmal et al., 2020). Chitin is further deacetylated to chitosan (CS), and its further hydrolysis can produce low-molecular-weight (MW) products, namely chitooligosaccharide (COS). Several chemicals (hydrogen peroxide (H_2_O_2_), ascorbic acid (AsA)/H_2_O_2_ redox pair, etc.), specific or non-specific enzymes, and physical techniques have been employed for COS production [1,2,3,4]. COS with low toxicity is biodegradable and biocompatible. Various bioactivities of COS such as antioxidant (AO), antimicrobial (AM), anticancer, and enzyme inhibitory activities have also been reported. Due to its unique bioactivities, its demand in the pharmaceutical industry has been increasing at constant rate [5]. COS has also been used in the food and nutrition arena with emphasis on food quality improvement and human health benefits. The presence of free amino or acetamido at C2 and hydroxyl (-OH) groups at the C3 and C6 positions favors COS for chemical modification [1]. COS has been modified via several reactions such as carboxylation, methylation, sulfation, phosphorylation, etc. to enhance the bio-efficacy and applications [6]. In addition, the grafting of COS with polyphenol (PPN) from plants has been known to drastically enhance its bioactivity.

PPNs in plants are present as secondary metabolites in diverse structures and possess at least an aromatic ring along with one or more -OH groups. PPNs can be categorized into major classes, namely phenolic acids, flavonoids, and stilbenoids based on the type of functional moiety and number of hydroxylated aromatic rings [7]. PPNs are also suitable for use as natural preservatives in the meat industry due to their excellent AO and AM activities. Furthermore, anti-inflammatory and anticancer activities of PPNs have been known to reduce cardiovascular disease, diabetes mellitus, and cancer (Sobhani et al., 2021). In addition, PPN can inhibit α-amylase, α-glucosidase, and lipase through nonspecific binding mechanisms [8,9]. Therefore, foods enriched in PPN have a profound effect on human health promotion or disease prevention, especially diabetes and its related complications such as cardiovascular diseases.

PPNs such as catechin (CAT), epigallocatechin gallate (EGCG), gallic acid (GAL), caffeic acid (CAF), ferulic acid (FER), proanthocyanidin, etc. have been successfully grafted onto CS to enhance its bioactivity and water solubility. However, limited studies have been conducted on the conjugation of COS with PPN. Eom, Senevirathne, and Kim [10] found increased AO activities when COS (MW: 3–5 kDa) was conjugated with different phenolic acids (hydroxybenzoic, protocatechuic, vanillic, syringic, p-coumaric, CAF, FER, and sinapic acids) as compared to COS alone. Similarly, squid-pen COS (MW: 76 kDa) conjugated with EGCG showed enhanced AO and AM activities as compared to COS alone [6]. COS grafted with GAL could inhibit the proliferation of AGS human gastric cancer cells [11]. In general, crosslinking chemicals such as (1-ethyl-3-(3-dimethylaminopropyl) carbodiimide, *N*,*N*′-dicyclohexylcarbodiimide, N hydroxysuccinimide, and hydroxybenzotriazole, etc. have been employed for conjugation of PPN on the COS backbone. However, chemical reaction is costly and toxic to humans, thus limiting its usage in food and pharma sectors [1,10,11]. As a consequence, alternative cheap, safe, and nontoxic methods could be used to graft different PPNs on COS. The redox pair method using AsA/H_2_O_2_ occurs at room temperature (RT), and the degradation of PPNs caused by oxidation can be avoided. To the best of our knowledge, very little information has been available on the preparation and characterization of COS conjugates using different types of PPNs by free-radical-mediated grafting methods. Furthermore, no information has been documented for the antidiabetic potential of COS, especially from shrimp-shell CS and its PPNs conjugates. These conjugates can be applied as the novel or alternative additives to retard the spoilage of perishable seafoods due to bacteria and to ensure the safety of products associated with some bacterial pathogen. Moreover, this study provides the novel information on the antidiabetic potential of COS and its PPN conjugates, which could serve as an effective antidiabetic agent. In addition, their antioxidant activity is related to the prevention of lipid oxidation in foods and prevention of several diseases associated with reactive oxygen species.

Thus, the study aimed to prepare and characterize the conjugates of COS with CAT, EGCG, GAL, CAF, and FER (natural phenolic antioxidants extractable from plants and the most common commercially available) at different levels. Firstly, an efficient method for the synthesis of COS-PPN conjugates was developed by using the AsA/H_2_O_2_ redox pair system. Then, the synthesized COS-PPN conjugates were subjected to detailed characterization using UV–vis, Fourier-transform infrared (FTIR), and nuclear magnetic resonance (NMR) spectroscopy to confirm the conjugation. Furthermore, AO and AM of COS conjugates were also tested. Finally, the α-amylase, α-glucosidase, and pancreatic lipase inhibitory activities of COS and its PPN conjugates were investigated.

## 2. Materials and Methods

### 2.1. Chemicals

Chitosan (MW: ~2100 kDa and DDA: 85%, determined using gel permeation chromatography and ^1^H-NMR, respectively) was purchased from Marine Bio Resources Co., Ltd., Samutsakhon, Thailand. Ascorbic acid and H_2_O_2_ (50%, *v*/*v*) were acquired from Loba Chemie Pvt. Ltd., Mumbai, India. GAL, FER, CAF, CAT, and EGCG were obtained from Xi’an Julong Bio-Tech Co., Ltd. (Xi’an, China). Further, α-amylase, α-glucosidase, porcine pancreatic lipase, rat intestinal acetone powder, acarbose, orlistat, 4-nitrophenyl α-D-glucopyranoside (PNP-glycoside), and 4-methylumbelliferyl oleate (4-MU) were purchased from Sigma-Aldrich (St. Louis, MO, USA). Chemicals used for AO activities were also acquired from Sigma-Aldrich (St. Louis, MO, USA). Microbial media were bought from HiMedia Laboratories, Mumbai, India.

### 2.2. Preparation of COS

COS was prepared through the AsA/H_2_O_2_ redox pair hydrolysis method [12]. Firstly, CS (1%, *w*/*v*) was dissolved in acetic acid (2%, *v*/*v*) overnight in an Erlenmeyer flask at room temperature, and then pH was adjusted to 5.0 using 6 M NaOH. Thereafter, redox-pair solution consisting of AsA and H_2_O_2_ at a molar ratio of 0.05/0.1 was mixed together and incubated at 40 °C for 15 min for -OH radical generation. Then, the aforementioned solution (2 mL) was added to 100 mL of CS solution to initiate hydrolysis. The mixture was shaken for 2 h at 60 °C with the aid of a shaker water bath (Memmert, D-91126, Schwabach, Germany). Thereafter, the mixture was allowed to cool in iced water, followed by the adjustment of pH to 7 using 6 M NaOH. The undissolved matter was removed from the mixture using a centrifuge (Himac CR22N, Tokyo, Japan) at 10,000× *g* for 15 min at 25 °C. The supernatant was collected and known as COS solution, which was further subjected to lyophilization (Scanvac Model Coolsafe 55, Coolsafe, Lynge, Denmark). COS powder was collected and packed in a zip-lock bag until use. COS had a degree of deacetylation (91%), degree of polymerization (2–8), and average molecular weight (0.7 kDa) as determined using ^1^H-NMR, MALDI-TOF-MS and gel permeation chromatography, respectively.

### 2.3. Preparation of COS-Polyphenol (COS-PPN) Conjugates

COS-PPN conjugates were prepared as described by Mittal, et al. [4]. Firstly, COS (1 g) was added to 100 mL of acetic acid (0.5%, *v*/*v*) and stirred overnight for complete solubilization. The pH of the COS solution was adjusted to 5.0 using 1 M NaOH in Erlenmeyer flasks. Then, 4 mL of H_2_O_2_ (1 M) containing 0.10 g AsA was incubated at 40 °C for 15 min to generate -OH radicals. The aforementioned solution was added to COS solution followed by incubation at room temperature for 60 min with constant stirring using a magnetic stirrer. Thereafter, GAL, FER, CAF, CAT, and EGCG were added into the mixture to obtain concentrations of 0.1, 0.2, 0.4, 0.6, 0.8, and 1% (*w*/*w* of COS) and further incubated at room temperature for 24 h in the dark. The obtained mixtures were dialyzed using a dialysis bag (MW cut-off: 500 Da) against 20 volumes of distilled water at 4 °C for 24 h to remove free PPN with two changes of dialysis medium. The dialysates were then lyophilized to obtain COS-PPN conjugate powders. COS conjugates produced by grafting of GAL, FER, CAF, CAT, and EGCG were defined as COS-GAL, COS-FER, COS-CAF, COS-CAT, and COS-EGCG, respectively.

### 2.4. Determination of Total Phenolic Content (TPC) and Conjugation Efficiency of Different COS-PPN Conjugates

The amount of GAL, FER, CAF, CAT, and EGCG in their respective conjugates was analyzed using Folin–Ciocalteu reagent (FCR) [4]. Briefly, all the conjugate powders were dissolved in deionized water to obtain a concentration of 1 mg/mL. Then, 0.1 mL of each sample was mixed with 0.75 mL of 0.2 N FCR and incubated for 5 min at room temperature in the dark. Thereafter, 0.75 mL of Na_2_CO_3_ (6%, *w*/*v*) was added and incubated for 1 h at room temperature. The absorbance of the mixtures was then read at 760 nm using a UV-visible spectrophotometer (UV-1800, Shimadzu, Kyoto, Japan). GAL, FER, CAF, CAT, and EGCG were used as a standard for their corresponding conjugate sample. TPC of COS-GAL, COS-FER, COS-CAF, COS-EGCG, and COS-CAT was expressed as mg GAL equivalent/g sample, mg FER equivalent/g sample, mg CAF equivalent/g sample, mg EGCG equivalent/g sample, and mg CAT equivalent/g sample, respectively. Conjugation efficiency was computed as per the following equation:Conjugation efficiency (%) = (Amount of polyphenol in COS-PPN conjugate)/(Initial amount of polyphenol used for conjugation) × 100(1)

For each PPN used, the resulting conjugate that possessed the highest conjugation efficiency was selected for further studies for comparison to COS.

### 2.5. Characterization of Selected COS-PPN Conjugates and COS

#### 2.5.1. UV-Visible, Fourier Transform Infrared (FTIR), and Proton (^1^H) Nuclear Magnetic Resonance (NMR) Spectra

UV-visible spectra were analyzed using a UV-visible spectrophotometer. Attenuated total reflection (ATR)-FTIR analysis was done using Bruker Invenio S FTIR spectrometer equipped with a A225/Q Platinum unit equipped with single reflection diamond crystal (Bruker OPTIK GmbH, Ettingen, Germany) [13]. ^1^H-NMR spectra were recorded on a 500 MHz Fourier-transform NMR spectrometer (Bruker AVANCE NEO 500 MHz, Ettingen, Germany) operated at 500.15 MHz.

#### 2.5.2. Antioxidant (AO) Activities

AO activities, including DPPH- and ABTS-radical scavenging activities (RSA), ferric reducing antioxidant power (FRAP), and metal chelating activity (MCA), were analyzed as per the methods explained by Benjakul, Kittiphattanabawon, Sumpavapol, et al. [14]. A sample concentration of 1 mg/mL was used for the analyses.

For DPPH-RSA, the sample (1.5 mL) was combined with 1.5 mL of 0.15 mM DPPH in 60 % ethanol. The mixture was mixed vigorously and allowed to stand at room temperature in the dark for 30 min. The absorbance of the resulting solution was measured at 517 nm using a spectrophotometer. A sample blank was prepared in the same manner except that the corresponding solvents were used instead of DPPH solution. A standard curve was prepared using Trolox in the range of 10–60 μM. The activity was calculated after subtraction of the sample blank.

For ABTS-RSA, the stock solutions included 7.4 mM ABTS solution and 2.6 mM potassium persulphate solution. The working solution was prepared by mixing the two stock solutions in equal quantities and allowing them to react for 12 h at room temperature in the dark. The solution was then diluted by mixing 1 mL of ABTS solution with 50 mL of methanol in order to obtain an absorbance of 1.1 ± 0.02 at 734 nm using the spectrophotometer. Fresh ABTS solution was prepared for each assay. The sample (150 μL) was mixed with 2850 μL of ABTS solution, and the mixture was left at room temperature for 2 h in the dark. The absorbance was then measured at 734 nm using a spectrophotometer. A sample blank was prepared in the same manner except that methanol was used instead of the ABTS solution.

To determine FRAP, stock solutions including 300 mM acetate buffer (pH 3.6), 10 mM TPTZ (2,4,6-tripyridyl-s-triazine) solution in 40 mM HCl, and 20 mM FeCl_3_·6H_2_O solution were prepared. A working solution was freshly prepared by mixing 25 mL of acetate buffer, 2.5 mL of TPTZ solution, and 2.5 mL of FeCl_3_·6H_2_O solution. The mixed solution was incubated at 37 °C for 30 min in a water bath (Memmert, D-91126, Schwabach, Germany) and was referred to as FRAP solution. A sample (150 μL) was mixed with 2850 μL of FRAP solution and kept for 30 min in the dark at room temperature. The ferrous tripyridyltriazine complex (colored product) was measured by reading the absorbance at 593 nm. A sample blank was prepared by substituting distilled water for FeCl_3_ in the FRAP solution. The standard curve was prepared using Trolox ranging from 50 to 600 μM.

For the MCA assay, the sample (940 μL) was mixed with 20 μL of 2 mM FeCl_2_ and 40 μL of 5 mM ferrozine. The reaction mixture was allowed to stand for 20 min at room temperature. The absorbance was then read at 562 nm. The blank was prepared in the same manner except that distilled water was used instead of the sample. For the sample blank, distilled water was substituted for the FeCl_2_ solution. The standard curve was prepared using EDTA ranging from 10 to 60 μM.

DPPH-RSA, ABTS-RSA, and FRAP were expressed as mmol Trolox equivalents (TE)/g sample, while MCA was reported as mmol EDTA equivalents (EE)/g sample. ORAC of the samples was determined as per the method of Chotphruethipong, Benjakul and Kijroongrojana [15] and expressed as mmol TE/g sample.

### 2.6. Antidiabetic Activity

Antidiabetic activity of different COS-PPN conjugates in comparison to COS was determined based on their potential to inhibit various digestive enzymes. In the preliminary study, lower sample concentration (0.5 mg/mL) showed no inhibitory effects against digestive enzymes (data not shown), whereas sample concentrations of 10 mg/mL were completely soluble in buffer solutions and showed high activity. Therefore, sample concentrations (1 to 10 mg/mL) were selected for analyses based on the preliminary study.

#### 2.6.1. α-Amylase Inhibitory Activity

Inhibitory activity of α-amylase was determined using soluble starch as a substrate as described by Awosika and Aluko [16] with some modifications. COS and the selected COS-PPN conjugates at different concentrations (1, 5, and 10 mg/mL) were dissolved in 0.02 67M sodium phosphate buffer (SPB; pH 6.9) containing 0.006 M NaCl. Then, the sample and α-amylase solution (100 μg/mL) were mixed (100 µL each) and incubated for 10 min at 37 °C. One hundred μL of starch solution (1%, *w*/*v* in SPB) was added after incubation and further incubated at 25 °C for 10 min. By adding 200 μL of dinitrosalicylic acid (DNSA) color reagent (96 mm DNSA, 2 M sodium potassium tartrate tetrahydrate, and 2 M NaOH), the reaction was stopped. Thereafter, the reaction mixture was immersed in a boiling water bath (95 °C) for 5 min and then allowed to cool to room temperature, followed by dilution using double-distilled water (3 mL). Finally, 200 μL of the reaction mixture was transferred to a 96-well microplate and read at 540 nm using a FLUOstar^®^ Omega microplate reader (BMG Labtech, Ortenberg, Germany) set at 25 °C. Buffer was used to replace enzyme and used as a blank. Acarbose, a pharmacological α-amylase inhibitor (0.1, 0.5, and 1 mg/mL), was used as a positive control. The control was prepared by using the same amount of double-distilled water instead of the sample. The α-amylase inhibitory activity (%) was calculated using the following equation:Inhibition (%) = [1 − ((As − Asb)/Ac)] × 100(2)
where Ac, As, and Asb denote the absorbance of the control, sample, and sample blank, respectively.

#### 2.6.2. α-Glucosidase Inhibitory Activity

The inhibitory activity on α-glucosidase was assayed as described by Awosika and Aluko [16]. Briefly, 500 mg of rat intestinal acetone powder was homogenized in 9 mL of NaCl solution (0.9%, *w*/*v*) and centrifuged at 12,000× *g* using Allegra^®^ 25R centrifuge (Beckman Coulter, CA, USA) for 30 min. The supernatant was used as the source of the α-glucosidase enzyme. COS and the selected COS-PPN conjugates at different concentrations (1, 5, and 10 mg/mL) were dissolved in 0.1 M SPB (pH 6.9). The solution (50 μL) was mixed with an equal volume (50 μL) of α-glucosidase enzyme (10 mg/mL) in a 96-well microtiter plate and incubated at 37 °C for 10 min. Thereafter, 100 μL of 5 mM 4-nitrophenyl α-D-glucopyranoside (PNP-glycoside) solution (dissolved in 0.1 M SPB, pH 6.9) was added to each well containing sample and enzyme solution and incubated for 30 min at 37 °C. The absorbance was read at 405 nm using a microplate reader. A blank (all reagents except the enzyme) was prepared. The α-glucosidase activity was determined by release of p-nitrophenol from PNP-glycoside, in which the absorbance was read at 405 nm. Acarbose (0.1, 0.5, and 1 mg/mL) was used as a positive control. The control was prepared in the same manner except double-distilled water was used to replace sample. The α-glucosidase inhibitory activity (%) was calculated using the equation shown below:Inhibition (%) = [1 − ((As − Asb)/Ac)] × 100(3)
where Ac, As, and Asb represent the absorbance of the control, sample, and sample blank, respectively.

#### 2.6.3. Lipase Inhibitory Activity

Lipase inhibitory activity was analyzed following the method detailed by Sultana, Alashi, Islam, et al. [17]. The release of 4-methylumbelliferone (4-MU) from 4-methylumbelliferyl oleate was measured to determine lipase activity. Twenty-five μL of sample dissolved at different concentrations (1, 5, and 10 mg/mL) in Tris buffer (13 mM Tris-HCl, 150 mM NaCl, and 1.3 mM CaCl2; pH 8.0) was mixed with 225 μL of 4-MU oleate solution (0.5 mM) and transferred to a 96-well microplate. The microplate was incubated for 15 min at 37 °C. Thereafter, 25 μL of pancreatic lipase solution (50 U/mL) was added to the mixture and incubated at 37 °C for 1 h. The amount of 4-MU released by the lipase was measured in a microplate reader at 400 nm after incubation. Orlistat, a pharmacological pancreatic lipase inhibitor (0.1, 0.5, and 1 mg/mL) was used as a positive control. The control was prepared by using double-distilled water instead of sample. The pancreatic lipase inhibitory activity (%) was calculated using the following equation:Inhibition (%) = [1 − ((As − Asb)/Ac)] × 100(4)
where Ac, As, and Asb indicate absorbance of the control, sample, and sample blank, respectively.

### 2.7. Antimicrobial Activity

#### 2.7.1. Zone of Inhibition (ZOI)

Bacterial strains including *Pseudomonas aeruginosa* PSU.SCB.16S.11, *Listeria monocytogenes* F2365, *Escherichia coli* DMST 4122, *Staphylococcus aureus* DMST 4547, and *Vibrio parahaemolyticus* PSU.SCB.16S.14 (gifted by the Food Safety Laboratory, Faculty of Agro-Industry, Prince of Songkla University, Hat Yai) were used to determine AM activity of COS and different selected COS-PPN conjugates. The ZOI was measured by the agar well diffusion method. Different samples (10 mg/mL) were dissolved in autoclaved water and sterilized using a 0.22 µm syringe filter. The lawn of bacteria was prepared using 1 mL of bacterial stock (10^6^ CFU/mL), which was transferred onto Mueller Hinton agar (MHA; 15 mL). After the agar solidified, wells were made (diameter of 0.2 mm) using a sterile tip. The samples (10 µL) were added to the wells. Thereafter, MHA plates were allowed to dry for 15 min in a laminar airflow chamber and then incubated overnight at 37 °C in an incubator (Memmert INB 200, GmbH + Co. KG, Schwabach, Germany). After the incubation, the ZOI was measured in millimeters (mm) using a vernier caliper with an accuracy of ±0.02 mm (Mitutoyo America Corporation, Aurora, IL, USA).

#### 2.7.2. Minimum Inhibitory Concentration (MIC) and Minimum Bactericidal Concentration (MBC)

Two-fold serial dilution was used for determination of the MIC and MBC of COS and the selected COS-PPN conjugates against the aforementioned bacterial strains as per the method of Mittal, Singh, and Benjakul [13]. The differences in absorbance after incubation was less than 0.02 at 630 nm between the sample and the negative control, indicating no bacterial growth. The lowest concentration of samples that inhibited the noticeable growth of bacteria in the microtiter plates after incubation was defined as MIC. MBC was recorded as the concentration of samples that completely inhibited the growth of bacteria after 10 μL of aliquots from wells were spotted on Tryptic soy agar (TSA) plates after being incubated for 24 h at 37 °C.

### 2.8. Statistical Analysis

All experiments were conducted in triplicate (*n* = 3) and a completely randomized design (CRD) was used. Analysis of variance and comparison of means via Duncan’s multiple range tests or *t*-test [18] were done using an SPSS-23 (SPSS Inc., Chicago, IL, USA).

## 3. Results and Discussion

### 3.1. Conjugation Efficiency of Various COS-PPN Conjugates

Conjugation efficiency of different PPNs at various concentrations on COS is given in Table 1. The degree of conjugation varied, with values of 35.94–29.59, 19.29–7.05, 19.60–7.93, 29.32–5.16, and 14.12–8.74% were obtained when GAL, FER, CAF, EGCG, and CAT, respectively, at various levels (1–10 mg/mL) were grafted. Generally, the -OH radical formed via AsA/H_2_O_2_ redox pair reaction abstracts hydrogen atoms from COS, resulting in the formation of the COS macro radical [1]. The -NH_2_ group at the C2 position and the -OH group at the C6 position of COS are considered to be major conjugation sites, while the -OH group at the C3 position is a less favorable position due to higher steric hindrance [19]. PPN acted as an acceptor of the COS macro radical, and the formation of a COS-PPN conjugate occurred. Overall, conjugation efficiency decreased with increasing concentration of the individual PPN (*p* < 0.05). The lower conjugation efficiency with increasing PPN levels was likely associated with the limited solubility of PPN at higher concentration in aqueous medium. As a result, the soluble PPN available for conjugation decreased. However, for COS-GAL and COS-CAT, higher conjugation efficiency was achieved when the concentration increased up to 4 mg/mL (*p* < 0.05). This was likely due to structural differences, thus determining their solubility and interaction with the functional groups of COS in different fashions. With further increases in GAL and CAT levels from 4 to 10 mg/mL, conjugation efficiency was decreased (*p* < 0.05). Among all the COS-PPN conjugates, the highest conjugation efficiencies (35.94 and 35.44%) were observed when 2 and 4 mg/mL GAL was used, respectively. However, no difference was noticed between both samples (*p* > 0.05). On the other hand, the lowest conjugation efficiency (5.16%) was obtained when 10 mg/mL EGCG was used for grafting with COS. For COS-FER, COS-CAF, and COS-EGCG, the conjugation efficiency decreased after PPN concentration was higher than 1 mg/mL (*p* < 0.05). In general, GAL had higher solubility in the aqueous solution as compared to other PPNs, more likely due to its smaller size and the presence of three -OH groups, which were able to form hydrogen bonding with COS effectively [20]. On the other hand, FER and CAF possess one and two -OH groups, respectively, on the benzene ring along with an aliphatic carbon chain. This limited their solubility in the aqueous medium, thus lowering the conjugation efficiency when compared to other PPNs. In addition, EGCG possessed a higher number of -OH groups, followed by CAT. As a result, their bulky structures limited the solubility in the aqueous medium and caused steric hindrance, thus restricting hydrogen bonding with COS molecules [21]. This was supported by the lowest conjugation efficiency of EGCG and CAT at 10 mg/mL. COS conjugated with GAL, FER, CAF, EGCG, and CAT at 2, 1, 1, 1, and 4 mg/mL, respectively, showed higher conjugation efficiency compared to the other treatments. Those aforementioned conjugates were selected for further study and defined as COS-GAL, COS-FER, COS-CAF, COS-EGCG, and COS-CAT, respectively.

### 3.2. Molecular Characteristics of the Selected COS-PPN Conjugates

#### 3.2.1. UV-vis Spectra

The UV-vis spectra of COS, different PPNs, and the selected COS-PPN conjugates are given in Figure 1A–F. COS showed a very small peak in the UV-vis spectral range. GAL showed two UV absorption peaks at 212 and 265 nm due to the π electron system of the benzene ring [22]. Benzene has six π electrons along with 12 electrons of the planar ring of C–C σ bonds. Its radical C–H σ bonds contain six electrons and are involved in forming the thermodynamically and kinetically stable benzene ring. The peak was shifted to a lower wavelength of 259 nm after grafting with GAL. FER showed three maximum absorption peaks at 215, 290, and 314 nm, whereas CAF exhibited three peaks at 215, 288, and 313 nm [22,23]. The n to π* and π to π* electronic transitions associated with the C=O group were responsible for the absorption at 290 or 288 and 215 nm, respectively. Nevertheless, the bands at 314 or 313 nm belong to the π to π* transitions of the aromatic moiety in FER and CAF [22]. For COS-FER, peaks were shifted to lower wavelength (i.e., 311 and 285 nm) reflecting the conjugation of FER onto COS. The peaks were shifted to 213, 285, and 311 nm in the COS-CAF conjugate, indicating a blue shift due to the incorporation of CAF onto COS via amide bonds [24]. Peaks at 269 and 273 nm were attributed to EGCG and CAT, respectively [25]. After grafting of EGCG and CAT on COS, peaks in the UV spectra were shifted to 273 and 274 nm, respectively, indicating a red shift. The red shift can be plausibly associated with less energy required for n to π* and π to π* transition due to the covalent linkage between PPN and COS [25]. The absorption peaks of COS-PPN conjugates were consistent with those of GAL, FER, CAF, EGCG, and CAT, reconfirming that all the PPNs used were successfully grafted onto COS.

#### 3.2.2. FTIR Spectra of COS, PPNs, and COS-PPN Conjugates

FTIR spectra of COS-GAL, COS-FER, COS-CAF, COS-CAT, and COS-EGCG in comparison with COS and corresponding PPNs are illustrated in Figure 2A–F. In COS, prominent bands between 3500 and 3250 cm^−1^ were observed, representing -OH and -NH stretching, respectively, as well as inter- and intramolecular hydrogen bonding [4]. The peaks associated with -OH and -NH overlapped due to inter- and intramolecular traces of water, which resulted in a wide band formation. The -CH_3_ asymmetrical stretching vibration appeared at 2930 cm^−1^ in COS. For amide-I, the bands were observed at 1647 cm^−1^ for COS, representing C=O stretching from the acetyl group [26]. The band at 1571 cm^−1^ represents amide-II, indicating N-H symmetric bending vibration and C-N stretching in COS [13]. The band associated with CH_2_ bending and deformation of CH_3_ appeared at 1412 cm^−1^ in COS. The symmetrical CH_3_ deformation and amide-III band appeared at 1371 and 1323 cm^−1^, respectively, in COS. Asymmetric stretching of the C–O–C bridge representing glycosidic linkage was found around 1150 cm^−1^ in COS [6]. The C–O–C asymmetrical stretching in the phase ring band was recorded at 1068 cm^−1^ in COS. The C–O asymmetrical stretching related to primary alcohol and secondary alcohol in COS appeared around 1030 cm^−1^ [27]. Moreover, the stretching of the glucosamine ring was detected between 890 and 899 cm^−1^ in COS.

GAL exhibited -OH stretching and hydrogen bonding between 3500 to 3200 cm^−1^ [28]. The bands of C=O stretch of conjugated acids appeared at wavenumbers of 1701 and 1616 cm^−1^. The COOH stretching and bending were observed at 1445 and 1244 cm^−1^, respectively, indicating a combination of O-C-C asymmetric stretching and -OH bend. The stretching of C=C in the aromatic ring was observed at 1445 cm^−1^. The in-plane bending of the -OH group appeared at 1365 cm^−1^. The bands associated with C-O/C-C stretching vibrations were attained at 1200–1300 cm^−1^. C-O stretching was evident at 1026 cm^−1^. As compared to COS, the intensity of the band representing amide-II (N-H symmetric bending vibration) of the primary -NH_2_ at 1571 cm^−1^ was decreased in the COS-GAL, indicating the change of the primary or secondary -NH_2_ due to the reactions at C2 on COS chains via the conjugation process. In addition, a new band associated with C=O stretching in esters appeared at 1730 cm^−1^ in COS-GAL, suggesting the formation of the ester bond between -OH groups of COS at C3 and/or C6 and carboxyl groups of GAL [29].

The bands in the FTIR spectra of both FER and CAF around 3400 cm^−1^ corresponded to vibrations of -OH stretching present in the benzene ring (phenols) [30]. The =CH stretching in the aromatic ring and unsaturated hydrocarbons were assigned at 3218 cm^−1^ for CAF. However, =CH stretching of the aromatic ring was found at 3000–2800 cm^−1^ in FER [31]. The band associated with -CH stretching of the acyclic chain was assigned at 2950 cm^−1^ for FER, while it was partially overlayed -OH stretching in the case of CAF [32]. The strong C=O stretching vibrations of the carboxylic group appeared at 1600–1700 cm^−1^. The stretching of the benzene ring and olefinic C-C stretching in the acyclic chain were noticeable at 1615 and 1534 cm^−1^, respectively. The strong intensity at 1440 cm^−1^ was attributed to in-plane bending of the -OH group in a carboxylic acid. For FER, the band associated with C–H vibration of the -CH_3_ group appeared at 1330 cm^−1^. However, the aforementioned band was absent in CAF. The bands of in-plane bending of olefinic and aromatic =CH bonds appeared at 1277 and 1207 cm^−1^, respectively. The bands around 1120 and 970 cm^−1^ were attributed to C-O stretch of the -OH group and carbon ring breathing in the benzene ring, respectively. The bands at 650 and 546 cm^−1^ indicate O-C=O and C-C=O bending of the carboxylic acid, respectively. In comparison to COS, FER grafted COS showed a novel peak at 1517 cm^−1^ (C=C aromatic ring) and augmented peak intensity of C-H stretching at 2926 cm^−1^ [30]. The result confirmed that FER was grafted onto COS. Similarly, the COS-CAF conjugate had a novel band found at 1537 cm^–1^, which could be assigned to C=C stretching of CAF moieties [33]. This confirmed that CAF was successfully grafted onto COS.

For EGCG and CAT spectra, the band between 3500 and 3200 cm^−1^ was dominant, which was more likely associated with -OH group vibration. However, the peak area was higher in EGCG due to the presence of a higher number of -OH groups as compared to CAT. The =CH stretching of the aromatic ring was assigned at 3000–2800 cm^−1^. The vibrations in the aromatic ring and C=C stretching were observed at 1600 cm^−1^ [6,34]. The absorption bands within 1200–1300 cm^−1^ corresponded to C-O or C-C stretching vibrations. Peaks within the range of 2900–3550 cm^−1^ were broader in the spectrum of COS-EGCG. The interaction of EGCG with the -OH group of COS was depicted via shift of band at 3434 cm^−1^ to the lower wavenumber (3418 cm^−1^) [35]. Moreover, shift of bands at 1641, 1560, and 1413 cm^−1^ to the lower wavenumbers, namely 1627, 1550, and 1409 cm^−1^, respectively, indicated strong interaction between COS and EGCG. The new peaks at 1148 and 1220 cm^−1^ in COS-EGCG were either due to -OH groups or the ether-cyclic nature of the EGCG [4]. The peak at 1040 cm^−1^ in COS-EGCG was more likely related to the stretching of the aromatic and aliphatic C–O bond of EGCG. Bands at 1148 cm^−1^ in the COS-EGCG conjugate were related to the C–O bonds in EGCG. Furthermore, the intensity was decreased for primary -NH_2_ band at 1550 cm^−1^, and the increased intensity of amide II and amide I bands were noticed in COS-EGCG. This was related to stretching vibrations of the C=C bond present in COS-EGCG. Thus, changes in the FTIR spectrum of COS-EGCG indicated the incorporation of EGCG at different binding sites onto COS.

The spectrum of COS-CAT showed the bands presenting NH stretching at 3300–3500 cm^−1^. This became broader due to enhanced stretching of aromatic and non-aromatic -OH groups. This reconfirmed the grafting of COS and CAT. The two adsorption bands corresponded to the PPN oligomers or polymers. The first peak at 1023 cm^−1^ represented the extended polymerization via the C-O stretching vibration, and the second peak at about 1053 cm^−1^ was associated with the PPN’s C-C linkages.

Overall, wavenumber shift was found in all COS-PPN conjugates compared to PPN alone. Peak intensities varied due to difference in TPC grafted to COS.

#### 3.2.3. ^1^H-NMR of COS, PPN, and COS-PPN Conjugates

The chemical structures of COS-GAL, COS-FER, COS-CAF, COS-CAT, and COS-EGCG conjugates were also further confirmed by ^1^H-NMR as given in Figure 3A–F. For the ^1^H-NMR spectrum of COS (Figure 3A), the signals at 4.92, 3.79–3.60, and 3.06 ppm were attributed to H1 (GlcN), H3–H6 (pyranose ring), and H2 (GlcN) proton, respectively. Moreover, protons attributed to the acetyl group appeared at 1.94 ppm. The acetyl protons (AcOH) at C6 were found at 2.09 ppm.

The ^1^H-NMR spectrum of GAL showed -OH protons at 8.83 and 9.20 ppm and a carboxyl group at 12.24 ppm (Figure 3B). The peak associated with the hydrogen atom of the aromatic ring appeared at 6.92 ppm [30]. As compared to COS, the COS-GAL conjugate showed a new peak related to phenyl protons of GAL at 6.9 ppm (Figure 3B).

FER showed proton signals at 3.81 ppm (-OCH_3_) and 6.35–6.38 ppm, 6.78–6.80 ppm, 7.07–7.09 ppm, 7.27 ppm, and 7.48–7.51 ppm, corresponding to the hydrogen atoms of the benzene ring [32] (Figure 3C). Protons of -OH and the acid group at 9.54 and 12.12 ppm, respectively, were obtained [36]. The grafting of FER onto COS led to the appearance of new strong signals at 6.38 to 7.39 ppm (methine protons of FER) [31] (Figure 3C).

CAF showed proton signal associated with an acid group at 12.10 ppm and -OH groups at 9.13 and 9.52 ppm (Figure 3D). The signals of methylene protons appeared at 7.48 and 6.35 ppm. The proton signals due to aromatic hydrogens were found at 6.38, 6.78, and 7.07 ppm. COS-CAF conjugate exhibited multiple peaks between 6.33–7.32 ppm, assigned to the methine (=CH-) protons of CAF [22].

EGCG generated seven proton signals in the ^1^H-NMR spectrum, including two singlet signals at 6.41 and 6.82 ppm, corresponding to the aromatic protons of the B- and D-ring, respectively (Figure 3E). Proton signals at 5.84 and 5.94 ppm were attributed to the A-ring. In addition, signals for aromatic protons of the C-ring were detected at 2.61, 4.91, and 5.32 ppm. This spectrum was in agreement with that of previous work [37]. When COS was conjugated with EGCG, a new peak at 6.99 ppm was noticed, which was attributed to aromatic protons of EGCG. These results proved the conjugation of EGCG on COS, in which weak downfield signals indicated that the degree of modification by EGCG-derived moieties was low.

Protons signals associated with -OH groups of the A- and B-ring of CAT appeared between 8.73 to 9.30 ppm (Figure 3F). The proton signal of a -OH group on the C-ring appeared at 4.97 ppm. The aromatic ring hydrogens of A-, B-, and C-rings were found at 5.94 and 5.84, 6.83 and 6.42, 4.97, 3.46, 2.50, and 2.65 ppm, respectively. The proton signals were in line with previous report [38]. The COS-CAT conjugate showed characteristic proton signals of COS and partial signals of CAT. Notably, no proton signal was observed for COS-CAT between 6.02–5.87 ppm (H–6 and H–8 of the CAT moiety), showing that H–6/H–8 of catechin (A-ring) was the conjugation position. Moreover, the presence of proton signals of the B-ring in COS-CAT indicated the grafting of catechin on COS. However, the aforementioned signals shifted to higher frequency. Based on FTIR and ^1^H NMR spectra, it could be concluded that the conjugation probably occurred between H–6 or H–8 of CAT (A-ring) and -OH of COS at C6.

### 3.3. Total Phenolic Content (TPC) and Antioxidant Activities of COS and COS-PPN Conjugates

TPC of COS-PPN conjugates is given in Table 2. In general, TPC indicates the reducing ability of the compound, in which a phosphowolframate–phophomolybdate complex is reduced by phenolics to blue products [4]. COS-CAT showed the highest TPC, followed by COS-GAL, COS-EGCG, COS-CAF, and COS-FER, respectively (*p* < 0.05). However, the TPC of COS-CAF and COS-FER did not differ (*p* > 0.05). The results were in agreement with FTIR and ^1^H-NMR spectra, in which successful conjugation of PPN on COS took place. Generally, TPC of PPN is dependent on the number of -OH groups. EGCG has the highest number (8) of -OH groups, while CAT, GAL, CAF, and FER consist of 6, 3, 2, and 1 -OH groups, respectively. Nevertheless, COS-EGCG (293.20 mg GAL/g sample) showed lower TPC as compared to COS-GAL and COS-CAT (*p* < 0.05). This might be associated with steric hindrance caused by the bulky EGCG molecule during conjugation, which eventually lowered the conjugation efficiency (Table 1). Conversely, COS-CAT had the highest TPC, followed by COS-GAL, although conjugation efficiency of the former was lower than the latter. This might be due to the higher amount of -OH group present in CAT. A similar result was observed in FTIR spectra, in which COS-CAT showed higher intensity in the -OH stretching region as compared to other samples (data not shown). This was plausibly associated with presence of a higher amount of -OH groups, which was related to higher TPC.

AO activity is considered as a crucial biological activity of COS-PPN conjugates. AO activities determined by different assays of various COS-PPN conjugates in comparison with COS are given in Table 2. After grafting, the AO activities of all the conjugates became higher than that of COS (*p* < 0.05), which is consistent with other research [6,22]. The AO activities of COS were mainly attributed to the presence of a protonated amino group at C2 of the pyranose ring, which can scavenge radicals [39]. Additionally, -OH groups at C3 and C6 of the pyranose ring of COS could function as a hydrogen donor to free radicals [40]. Moreover, the increased AO activities were more likely associated with an increasing number of -OH groups from the PPNs. GAL is a trihydroxybenzoic acid, the -OH at para and methoxy groups at the meta position of the benzene ring in FER determined AO activity. Two -OH groups at the para and meta position of the benzene ring in CAF plausibly augmented AO activities of COS conjugates. EGCG possessed eight -OH groups, out of those -OH groups, C3, C4, and C5 and the gallate moiety at the 3′ position in the C-ring are considered to be good electron donors. Moreover, the electron delocalization ability of EGCG might contribute to its AO activity.

For DPPH-RSA, COS-CAT had the highest activity, followed by COS-CAF and COS-GAL (*p* < 0.05). However, no difference in DPPH-RSA was noticed between COS-CAF and COS-GAL (*p* > 0.05). The lowest DPPH-RSA was obtained for COS-EGCG (*p* < 0.05). A similar trend was noticed for ABTS-RSA, in which the ABTS-RSA of the COS-PPN conjugate was in descending order: COS-CAT > COS-GAL > COS-CAF > COS > COS-FER > COS-EGCG > COS (*p* < 0.05). The result was governed by the highest TPC of COS-CAT, whereas steric constraints due to the bulky structure of EGCG limited the activity of the COS-EGCG conjugate. Generally, the AO activity of any compound is based on its ability to transfer hydrogen atoms or single electrons in a particular reaction system. ABTS-RSA has been used to measure hydrophilic systems, while DPPH-RSA is suitable for lipophilic systems. In general, ionization potential and proton dissociation energy of the antioxidants determined AO activity [41]. AO activities of COS and COS-PPN conjugate determined using DPPH-RSA and ABTS-RSA were mainly due to their ability to provide hydrogen atoms and show single electron transfer [1].

Unlike DPPH-RSA and ABTS-RSA, COS-EGCG showed higher FRAP and MCA among all samples except COS-CAT, which still possessed the highest activity (*p* < 0.05). The reducing power of antioxidants has been normally estimated by FRAP assay. COS and its PPN conjugates effectively reduced TPTZ-Fe^3+^ to TPTZ-Fe^2+^ complex via electron transfer. Fe^2+^ is a potent pro-oxidant in food systems. Thus, chelation of Fe^2+^ is associated with the inhibition of lipid oxidation of both food and oxidative-stress-related diseases [12]. COS might chelate metal ions via lone pair electrons [42]. Furthermore, the donation of lone pair electrons of -OH groups from PPNs contributes to metal ion chelation and its reduction from the highly reactive oxidized form to the less reactive form [13]. MCA is also influenced by ionic diameter, ring size, and conformation of the chelator, which further affect the stability of the chelator-metal complex [43]. The AO activity of PPNs is therefore related to the number and distribution of -OH groups, as well as the structure of PPNs [7].

The ORAC of COS and COS-PPN conjugates is given in Table 2. The ORAC has been used for the determination of the AO potential of the compound through inhibition of peroxyl-radical-induced oxidation. The ORAC has been used to indicate the radical chain-breaking potential of antioxidants by hydrogen atom transfer [44]. Generally, the peroxyl radical reacts with a fluorescent probe to form a non-fluorescent product, which can be quantitated easily by fluorescence over time. The principle of this assay is based on the intensity of fluorescent molecules, such as fluorescein, which decreases in time under the reproducible and constant flux of peroxyl radicals generated from the thermal decomposition of 2,2-azobis(2-amidino-propane) dihydrochloride (AAPH) in an aqueous buffer [44]. In the absence of antioxidants, the blank or control showed a sharp decrease in fluorescent intensity. Similar to AO activities, COS-CAT (80.27 mmol TE/g sample) showed the highest inhibition of peroxyl-radical-induced oxidation, likely due to its high TPC value, followed by COS-GAL, COS-EGCG, COS-FER, and COS-CAF (*p* < 0.05). However, no difference was observed in ORAC between COS-GAL and COS-CAT (*p* > 0.05). A similar ORAC was also obtained between COS-CAF and COS-FER (*p* > 0.05).

### 3.4. In Vitro Inhibitory Activities of COS and Its PPN Conjugates against α-Amylase, α-Glucosidase, and Pancreatic Lipase

Inhibition of α-amylase activity by various COS-PPN conjugates in comparison with COS at different concentrations is shown in Figure 4A; α-amylase is known to release oligosaccharides after the digestion of dietary starch, which further breaks down into the glucose that is rapidly absorbed by the body [16]. Thus, α-amylase inhibition is regarded as an effective strategy for managing diabetes. All COS-PPN conjugates showed a higher α-amylase inhibitory effect than COS. COS-CAT had the highest inhibitory potential irrespective of concentration used (*p* < 0.05). The α-amylase inhibitory activity of all the samples increased with augmenting concentrations (*p* < 0.05). However, no difference in α-amylase inhibitory activity of COS was observed when concentration increased from 1 to 10 mg/mL (*p* > 0.05). The highest α-amylase inhibitory activities for COS, COS-GAL, COS-FER, COS-CAF, COS-EGCG, and COS-CAT at 10 mg/mL were found to be 19.65, 52.98, 17.05, 37.31, 51.83, and 87.92%, respectively. Nevertheless, acarbose (1 mg/mL), which is a positive control, showed the highest α-amylase inhibitory activity of 93.04% (*p* < 0.05) as compared to all COS-PPN samples. The higher concentration of COS-PPN conjugates might effectively inhibit enzyme activity through hydrogen bonding or ionic interaction with amino acid residues of active sites of an enzyme, which changes the structure of the enzyme [45]. At the same concentration, α-amylase inhibitory activity of COS-PPN conjugates was in descending order: COS-CAT > COS-GAL > COS-EGCG > COS-FER > COS-CAF > COS (*p* < 0.05). The differences in inhibitory activity of various PPN conjugates might be due to structural variation and the number of -OH groups in both COS and PPN. CAT might interact with the side chains and form hydrogen bonds with Asp 197 and Asn 298 residues of the active site of the enzyme and may form π-π interactions between stacked, delocalized ring systems of the PPN and the indole ring of Trp 62 in α-amylase [46]. The higher inhibitory activity of COS-CAT is most probably due to high binding affinity with the active site of the enzyme. For COS-GAL, it plausibly formed hydrogen bonding and ionic interaction with Asp 197, Arg 195, and His 299 residues of the enzyme’s active site [47]. Similarly, hydrogen bonding between the -OH groups in the galloyl moiety and other position of EGCG and α-amylase residues (e.g., Glu 233) might take place. Moreover, π-conjugation force between the benzene ring in the galloyl moiety and benzene ring of aromatic amino acid in the active sites of enzyme (e.g., Tyr 62, Trp 58, and Trp 59) might occur [48]. Moreover, the -OH groups at 5, 6, and 7 positions in ring A and 4′ position in ring B advocate the inhibition effect of flavonoids. In addition, the 2, 3-double bonds in ring C also play a crucial role in the inhibition of α-amylase by flavonoids (CAT and EGCG) [48]. For COS-FER and COS-CAF, π-π T-shaped orientation exerted between the benzene ring of hydroxycinnamic acids (FER and CAF) and Try 3 and Tyr 151 of α-amylase could suppress the activity of α-amylase. Additionally, FER inhibited α-amylase via noncompetitive binding with Gly 334, Phe 335, Thr 6, Thr 11, Arg 398, and Pro 4, which altered the active site and inhibit the substrate binding [48,49].

The α-glucosidase is membrane-bound and is involved in starch digestion to produce glucose via breakdown of oligo- and disaccharides [17]. As depicted in Figure 4B, COS and COS-PPN conjugates exhibited a concentration-dependent α-glucosidase inhibitory activity. Similar to α-amylase, COS-CAT showed the highest enzyme inhibitory activity in a concentration-dependent manner (*p* < 0.05). Moreover, COS-FER showed lower α-glucosidase inhibitory activity than COS irrespective of the concentration used (*p* < 0.05). The binding affinity of the compound could be strongly influenced by amino residues of the active site in the enzyme. Similar to α-amylase inhibitory activity, COS-PPN conjugates had lower α-glucosidase inhibitory activity than acarbose (60.32% at 1 mg/mL) (*p* < 0.05). At the concentration of 10 mg/mL, the α-glucosidase inhibitory activities of COS, COS-GAL, COS-FER, COS-CAF, COS-EGCG, and COS-CAT were 18.61, 13.69, 14.76, 23.15, 26.06, and 44.48%, respectively. The inhibitory activity on α-glucosidase of different conjugates was in descending order: COS-CAT > COS-EGCG > COS-CAF > COS > COS-FER=COS-GAL (*p* < 0.05). The CAT-α-glucosidase complex involved hydrogen bonding between CAT and α-glucosidase residues (Arg 411, Asp 382, and Ala 59), which is plausibly associated with enzyme inhibition [47]. In addition, CAT was able to form a hydrogen bond with Asp 198, Asp 324, Asn 324, and Tyr 63 residues of α-glucosidase [50]. The lower intramolecular energy of CAT than EGCG and other PPN enhanced the stability of the CAT-α-glucosidase binary complex, which was mainly due to the higher binding affinity of CAT to α-glucosidase than other PPN [51]. Moreover, the trans-conformation of CAT favored hydrogen bonding between the 3-OH of CAT and essential amino acid residues (Tyr 72) in the α-glucosidase active site, thus contributing to higher efficacy of enzyme inhibition [52]. The presence of a galloyl group at position 3 of the C-ring and the -OH group at position 4 on the B-ring in EGCG contributed to α-glucosidase inhibitory activity. PPN such as FER mainly bind to amino acid residues (Asp 215, Glu 277, Gln 279, Arg 315, Glu 411, Tyr 158, Phe 159, and Phe 178) of the enzyme, which forms hydrogen bonding or π-π T-shaped interactions with other amino acid residues of α-glucosidase [53]. In addition, hydroxycinnamic acids (FER and CAF) inhibit α-glucosidase via noncompetitive inhibition [54]. For COS-GAL, GAL has been known to alter the enzyme structure via binding with amino acid residues (Asp 326, Asn 326, Asp 324, and Arg 197) of α-glucosidase [51]. Moreover, hydrogen bonding between GAL and Ser 240 and Ser 241 of α-glucosidase could inhibit its activity [55].

Pancreatic lipase is responsible for the intestinal digestion of dietary triacylglycerols, a major source of excess calorie intake. Enhanced absorption of lipids in the pancreas may activate pancreatic insulin-producing β cells, which cause type I or type II diabetes [56]. Therefore, the suppression or delay of lipid digestion and absorption through the inhibition of lipase activity could prevent the pancreas from producing β cells at a higher amount, which enables production of normal level of insulin and helps with the management of obese patients with diabetes. Unlike α-amylase and α-glucosidase inhibition, more than 80% reduction in lipase activity was achieved irrespective of sample and concentration used (except for COS-FER and COS-CAF at a concentration of 1 mg/mL, which showed inhibitory activity less than 80%) (Figure 4C). All samples effectively inhibited lipase activity in a dose-dependent manner. TPC values of each sample showed a high positive correlation with lipase inhibition [57]. Similar to α-amylase and α-glucosidase inhibitory activities, COS-CAT showed higher potential toward inhibition of lipase activity than the other conjugates (*p* < 0.05) expect for COS-CAF, which showed similar inhibitory activity to COS-CAT at levels of 5 and 10 mg/mL. Orlistat (1 mg/mL), a positive control for lipase inhibitory activity, showed 96.54% inhibition, which was similar to those of COS-CAT and COS-CAF at the same concentration (*p* > 0.05). In general, all PPNs inhibited pancreatic lipase in a noncompetitive manner via binding Asn 263 and Asp 206—which form a pocket adjacent to the active site—and altering its structure. Moreover, protonation of His 264 via a galloyl moiety could lead to the inactivation of lipase [50,58,59]. Thus, PPN grafted on COS played a major role in inhibition of lipase, and their effectiveness depended on the concentrations used.

### 3.5. Antimicrobial Activity of COS and COS-PPN Conjugates

AM activities expressed as MIC, MBC, and ZOI (mm) of COS and COS-PPN conjugates against five different Gram-negative and Gram-positive bacteria, including pathogenic and spoilage strains, are presented in Table 3. COS had higher MIC and MBC values than all COS-PPN conjugates, indicating that conjugation of PPNs on COS effectively increased the AM potential of COS. Similar results were obtained when squid-pen COS was conjugated with EGCG, which showed augmented AM activity [6]. Among all the conjugates, COS-CAT showed the lowest MIC and MBC values, indicating higher AM activity irrespective of tested bacteria. The result was in agreement with the highest TPC and AO activities of COS-CAT (Table 2). The MBC/MIC ratio can be used to determine the antibacterial potential of substances. COS, COS-GAL, and COS-FER showed an MBC/MIC ratio of one, regardless of the tested bacteria. Moreover, an MBC/MIC ratio of one was observed for *Listeria monocytogens* treated with COS-CAF and COS-CAT and *Pseudomonas aerugonisa and Staphylococcus auerus* treated with COS-CAF and COS-CAT, respectively. COS-EGCG had an MBC/MIC ratio of two against all tested bacteria, whereas COS-CAT showed an MBC/MIC ratio of two for *Pseudomonas aerugonisa*, *Vibrio parahaemolyticus*, and *Escherichia coli*. Generally, an antimicrobial agent having an MBC/MIC ratio ≤2 is considered as ‘bacteriostatic’. Thus, all conjugates along with COS possessed bacteriostatic properties [6].

It was observed that Gram-negative bacteria have the hydrophilic thin outer membrane, which consists of lipopolysaccharides. This made them more prone to cellular lysis via COS and its PPN conjugates [60]. The result was correlated with higher ZOI for Gram-negative bacteria including *Pseudomonas aerugonisa* (ZOI: 5.62–12.38 mm), *Vibrio parahaemolyticus* (ZOI: 5.32–13.65 mm), and *Escherichia coli* (ZOI: 4.23–9.45 mm) than Gram-positive bacteria. Gram-positive bacteria were generally more resistant than their Gram-negative counterparts due to the presence of a thick peptidoglycan layer [18]. Generally, the AM activity of COS and PPNs was associated with the breakdown of the bacterial cell wall through electrostatic interaction due to amino and -OH groups. In addition, changes in microbial DNA, mRNA, and protein synthesis via diffused COS and PPN could lead to cell death [1]. Thus, COS-PPN, especially COS-CAT, are a potential antimicrobial agent able to retard or prevent the growth of both spoilage and pathogenic bacteria.

## 4. Conclusions

COS-PPN conjugates were prepared using COS and GAL, FER, CAF, EGCG, and CAT at various concentrations via the free radical grafting method. FTIR and ^1^H-NMR analyses confirmed the successful grafting of PPNs on COS. After grafting of various PPNs on COS, antioxidant activities, including DPPH-RSA, ABTS-RSA, FRAP, MCA, and ORAC, were drastically increased, while COS-CAT had the highest activity. COS-CAT also showed the highest antimicrobial activity toward both Gram-positive and Gram-negative bacteria. Thus, COS-CAT can be used in various foods, especially seafoods, which are susceptible to microbial spoilage and lipid oxidation. Moreover, it could be incorporated in active packaging, with the additional function of serving as the additives. Moreover, COS-CAT showed higher α-amylase, α-glucosidase, and pancreatic lipase activity inhibitory effects than other conjugates and COS. Therefore, it can be used as an antidiabetic agent for diabetes control as a functional ingredient, with further benefits as an antioxidant to prevent several diseases.

## Figures and Tables

**Figure 1 foods-11-00920-f001:**
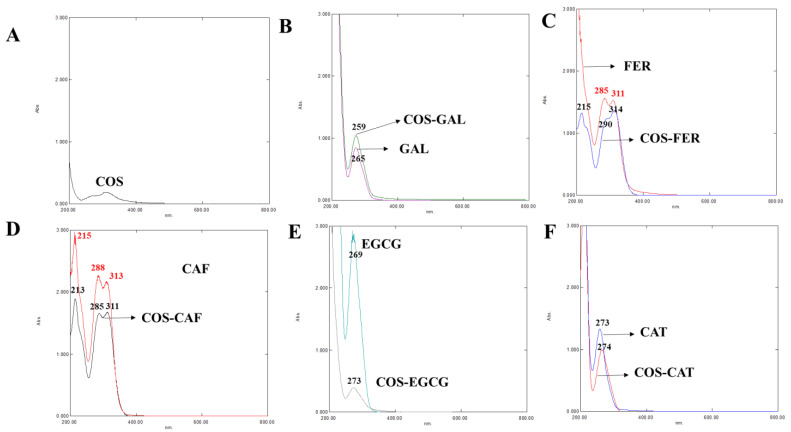
UV spectra of COS (**A**), GAL and COS-GAL (**B**), FER and COS-FER (**C**), CAF and COS-CAF (**D**), EGCG and COS-EGCG (**E**), and CAT and COS-CAT (**F**): COS, chitooligosaccharide; GAL, gallic acid; FER, ferulic acid; CAF, caffeic acid; EGCG, epigallocatechin gallate; CAT, catechin. The black and red numbers on the peaks indicate their wavelength.

**Figure 2 foods-11-00920-f002:**
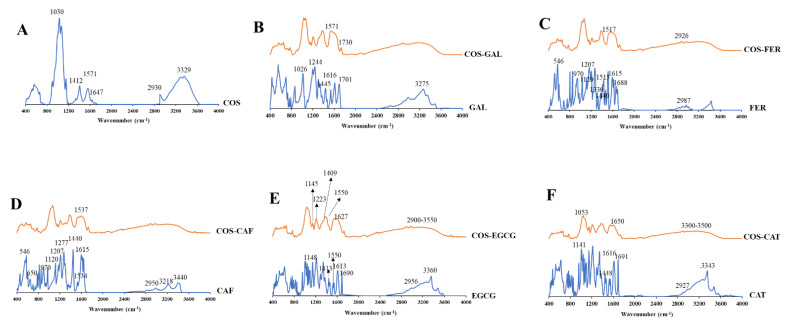
FTIR spectra of COS (**A**), GAL and COS-GAL (**B**), FER and COS-FER (**C**), CAF and COS-CAF (**D**), EGCG and COS-EGCG (**E**), and CAT and COS-CAT (**F**): COS, chitooligosaccharide; GAL, gallic acid; FER, ferulic acid; CAF, caffeic acid; EGCG, epigallocatechin gallate; CAT, catechin.

**Figure 3 foods-11-00920-f003:**
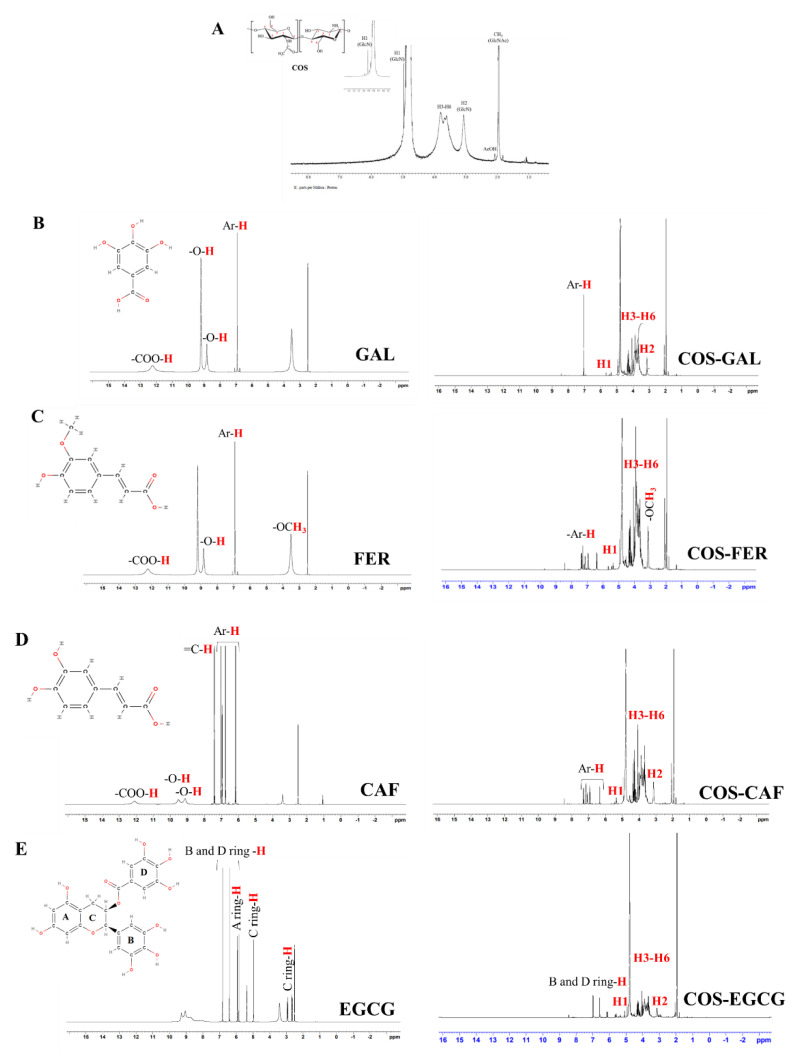
1H-NMR spectra of COS (**A**), GAL and COS-GAL (**B**), FER and COS-FER (**C**), CAF and COS-CAF (**D**), EGCG and COS€CG (**E**), and CAT and COS-CAT (**F**): COS, chitooligosaccharide; GAL, gallic acid; FER, ferulic acid; CAF, caffeic acid; EGCG, epigallocatechin gallate; CAT, catechin.

**Figure 4 foods-11-00920-f004:**
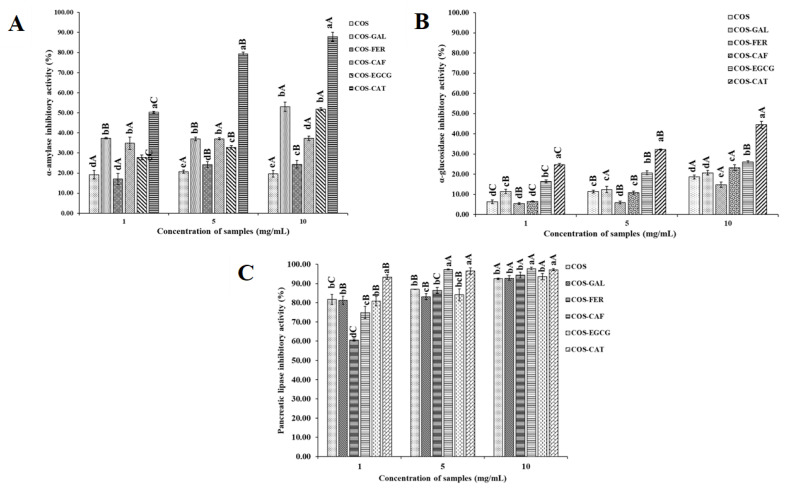
α-amylase (**A**), α-glucosidase (**B**), and pancreatic lipase (**C**) inhibitory activities of COS, COS-GAL, COS-FER, COS-CAF, COS-EGCG, and COS-CAT. Different uppercase superscripts within the same sample indicate significant differences (*p* < 0.05); different lowercase superscripts within the same concentration indicate significant differences (*p* < 0.05): COS, chitooligosaccharide; GAL, gallic acid; FER, ferulic acid; CAF, caffeic acid; EGCG, epigallocatechin gallate; CAT, catechin.

**Table 1 foods-11-00920-t001:** Conjugation efficiency of different phenolic compounds at various concentrations conjugated on chitooligosaccharide.

Concentration of Phenolic Compound Used (mg/mL)	Conjugation Efficiency (%)				
	Gallic Acid	Ferulic Acid	Caffeic Acid	Epigallocatechin Gallate	Catechin
1	29.56 ± 0.36 ^d^	19.29 ± 0.34 ^a^	19.60 ± 1.02 ^a^	29.32 ± 0.96 ^a^	14.12 ± 1.03 ^b^
2	35.94 ± 0.65 ^a^	13.10 ± 0.42 ^b^	8.20 ± 0.30 ^b^	17.59 ± 0.51 ^b^	14.52 ± 0.37 ^b^
4	35.44 ± 0.18 ^a^	9.30 ± 0.07 ^c^	8.10 ± 0.46 ^b^	9.56 ± 0.33 ^c^	17.22 ± 0.27 ^a^
6	33.03 ± 0.24 ^b^	9.83 ± 0.04 ^d^	7.93 ± 0.78 ^b^	9.05 ± 0.42 ^c^	10.98 ± 0.85 ^c^
8	32.48 ± 0.18 ^b^	7.05 ± 0.05 ^e^	8.02 ± 0.10 ^b^	8.73 ± 0.55 ^c^	9.39 ± 0.43 ^d^
10	31.27 ± 0.12 ^c^	7.93 ± 0.02 ^f^	8.54 ± 0.10 ^b^	5.16 ± 0.10 ^d^	8.74 ± 0.58 ^d^

Data are presented as mean ± SD (*n* = 3). Different lowercase superscripts in the same column indicate significant differences (*p* < 0.05).

**Table 2 foods-11-00920-t002:** Total phenolic content and antioxidant activities of chitooligosaccharide and its conjugate with different phenolic compounds.

	TPC ^#^	DPPH-RSA *	ABTS-RSA *	FRAP *	MCA **	ORAC *
COS	-	21.27 ± 0.24 ^e^	33.49 ± 0.12 ^f^	17.08 ± 0.57 ^f^	4.01 ± 0.35 ^d^	12.58 ± 1.25 ^d^
COS-GAL	518.89 ± 3.96 ^b^	116.04 ± 1.51 ^b^	357.70 ± 2.95 ^b^	170.04 ± 1.72 ^c^	8.27 ± 0.88 ^c^	78.58 ± 1.21 ^a^
COS-FER	192.86 ± 3.39 ^d^	102.64 ± 1.05 ^c^	166.43 ± 2.74 ^d^	115.14 ± 3.00 ^e^	10.11 ± 1.70 ^c^	30.54 ± 0.56 ^c^
COS-CAF	196.00 ± 2.16 ^d^	114.23 ± 1.57 ^b^	237.20 ± 2.23 ^c^	131.14 ± 3.13 ^d^	8.38 ± 0.90 ^c^	28.66 ± 1.45 ^c^
COS-EGCG	293.20 ± 5.92 ^c^	73.12 ± 1.71 ^d^	67.20 ± 2.88 ^e^	278.36 ± 0.11 ^b^	15.27 ± 1.42 ^b^	68.27 ± 0.99 ^b^
COS-CAT	688.79 ± 3.91 ^a^	123.22 ± 1.73 ^a^	365.83 ± 3.19 ^a^	409.93 ± 2.97 ^a^	20.13 ± 0.59 ^a^	80.27 ± 0.79 ^a^

Data are represented as mean ± SD (*n* = 3). Different lowercase superscripts in the same column indicate significant differences (*p* < 0.05): COS, chitooligosaccharides; GAL, gallic acid; FER, ferulic acid; CAF, caffeic acid; EGCG, epigallocatechin gallate; CAT, catechin; TPC, total phenolic content; DPPH-RSA, 2,2 diphenyl-1-picrylhydrazyl-radical scavenging activity; ABTS-RSA, 2,2-azinobis-(3-ethylbenzothiazoline-6-sulfonic acid) diammonium salt-radical scavenging activity; FRAP, ferric reducing antioxidant power; MCA, metal chelating activity; ORAC, oxygen radical absorbance capacity. ^#^ mg gallic acid equivalent/g sample; * mmol Trolox equivalent/g sample; ** mmol EDTA equivalent/g sample.

**Table 3 foods-11-00920-t003:** Antimicrobial activities of chitooligosaccharide and its conjugates with different phenolic compounds.

Parameters	Samples	*Pseudomonas aeruginosa*	*Vibrio parahaemolyticus*	*Escherichia coli*	*Listeria monocytogenes*	*Staphylococcus aureus*
MIC (µg/mL)	COS	1024	1024	2048	2048	2048
	COS-GAL	512	512	512	1024	1024
	COS-FER	512	256	512	512	512
	COS-CAF	256	256	512	512	512
	COS-EGCG	64	64	256	128	128
	COS-CAT	32	32	128	128	256
MBC (µg/mL)	COS	1024	1024	2048	2048	2048
	COS-GAL	512	512	512	1024	1024
	COS-FER	512	256	512	512	512
	COS-CAF	256	512	1024	512	1024
	COS-EGCG	128	128	512	256	256
	COS-CAT	64	64	256	128	256
Zone of inhibition (mm)	COS	5.62 ± 0.07 ^f^	5.32 ± 0.10 ^f^	4.23 ± 0.02 ^f^	4.31 ± 0.07 ^f^	4.32 ± 0.02 ^f^
	COS-GAL	7.54 ± 0.10 ^d^	6.23 ± 0.08 ^e^	5.32 ± 0.10 ^e^	5.45 ± 0.03 ^e^	4.72 ± 0.05 ^e^
	COS-FER	6.45 ± 0.03 ^e^	7.58 ± 0.10 ^d^	6.18 ± 0.02 ^d^	7.11 ± 0.05 ^d^	7.55 ± 0.01 ^d^
	COS-CAF	7.69 ± 0.14 ^c^	7.77 ± 0.07 ^c^	6.59 ± 0.02 ^c^	7.50 ± 0.14 ^c^	7.94 ± 0.07 ^c^
	COS-EGCG	8.54 ± 0.10 ^b^	8.83 ± 0.02 ^b^	8.58 ± 0.10 ^b^	8.44 ± 0.02 ^b^	8.79 ± 0.08 ^b^
	COS-CAT	12.38 ± 0.10 ^a^	13.65 ± 0.01 ^a^	9.45 ± 0.02 ^a^	8.86 ± 0.02 ^a^	10.47 ± 0.02 ^a^

Data are presented as mean ± SD (*n* = 3). Different lowercase superscripts in the same column indicate significant differences (*p* < 0.05): COS, chitooligosaccharides; GAL, gallic acid; FER, ferulic acid; CAF, caffeic acid; EGCG, epigallocatechin gallate; CAT, catechin; MIC, minimum inhibitory concentration; MBC, minimum bactericidal concentration.

## Data Availability

The data are not shared.

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
