# Peer review of "Chitooligosaccharide Conjugates Prepared Using Several Phenolic Compounds via Ascorbic Acid/H2O2 Free Radical Grafting: Characteristics, Antioxidant, Antidiabetic, and Antimicrobial Activities"

_foods, 2022, doi:10.3390/foods11070920_

Round 1

Reviewer 1 Report

In this manuscript, chitooligosaccharides were grafted with different polyphenol compounds, and their properties and activities were studied. Chitooligosaccharide-polyphenol conjugates with high activity were obtained. The research makes good sense. There are many experimental methods without reference, such as Line 98, 162…. Some infrared results are not discussed accurately and need to be checked carefully. Further discussion on the relationship between activity change and structure before and after grafting should be added. What is the solubility of the samples, and how to ensure that they are completely dissolved during the activity test.

Author Response

Reviewer 1

In this manuscript, chitooligosaccharides were grafted with different polyphenol compounds, and their properties and activities were studied. Chitooligosaccharide-polyphenol conjugates with high activity were obtained. The research makes good sense. There are many experimental methods without reference, such as Line 98, 162…. Some infrared results are not discussed accurately and need to be checked carefully. Further discussion on the relationship between activity change and structure before and after grafting should be added. What is the solubility of the samples, and how to ensure that they are completely dissolved during the activity test.

******Thank you so much for your valuable time spent on our manuscript. All suggestions have been taken into consideration for the improvement of the quality and clarity of our manuscript. All queries have been responded and corrcetions have been made as suggested by the reviewer as highlighted in yellow color

******Thank you for your valuable comment. The reference associated with the preparation of chitooligosaccharide has been added in the text. Please see line 115 and the reference list. Moreover, line 213 (previously line 162) associated with the section of the antidiabetic activity, which was divided further into three subsections for three different enzyme inhibitory activities. The references for the assays for those activities had already been cited in the text. Please see lines 222, 242, and 262.

******The results related to infrared spectroscopy have been checked carefully, especially for the Important peaks in spectrum. The results have been cross-checked and validated with previous reports. The discussion was made properly.

 ******Overall, COS-PPN conjugates showed higher bioactivities than COS, irrespective of polyphenol used. The enhanced bioactivities of COS-PPN conjugate are associated with an increased total phenolic content (TPC). The conjugation of polyphenols on COS was confirmed using FTIR and 1H-NMR spectra, which had been already discussed in the text. Moreover, increased TPC was in agreement with FTIR spectra, in which changes in intensity were observed among COS-PPN conjugates in -OH stretching region. The additonal discussion has been added to the text for clarity. Please see line 509-510 and 518-521.

******During the preliminary study, all samples at different concentrations (0.5 to 10 mg/mL) were tested for their solubility in water and assay buffer. All the samples were completely soluble up to 10 mg/mL in both water and buffer. However, some samples such as COS and COS-EGCG conjugate were not completely dissolved at concentrations above 10 mg/mL. Therefore, the authors used the samples at concentrations up to 10 mg/mL for the analyses. The concentration of sample used in each analysis was mentioned in the material and methods section for a better understanding of the reader.

Reviewer 2 Report

In this manuscript of “Chitooligosaccharide conjugates prepared using several phenolic compounds via ascorbic acid/H2O2 free radical grafting: characteristics, antioxidant, antidiabetic, and antimicrobial activities”, many parts of the manuscript are designed logically and systematically. However, there are still some questions for you to answer. And few refinements are needed to be done. The following are the questions and comments of this manuscript:

  1. Why detect α-amylase inhibitory activity at 25℃, but detect α-glucosidase inhibitory activity and lipase inhibitory activity at 37℃? What is the choose principle?
  2. Please provide detailed experimental steps about DPPH-, ABTS-, RSA, FRAP and MCA.
  3. How did the author decide on the different concentrations (10, 5, and 1 mg/mL)?
  4. I could not figure out the purpose of this study. Why did the author put these two different materials in together? What's the difference? Whether they address the AO, anti-diabetes and AM symptoms through a similar or distinguished approach? The introduction does not explain the choice of raw materials very well.
  5. What's the advantage or novelty of this system compared with other chitooligosaccharide conjugates systems?

Author Response

In this manuscript of “Chitooligosaccharide conjugates prepared using several phenolic compounds via ascorbic acid/H2O2 free radical grafting: characteristics, antioxidant, antidiabetic, and antimicrobial activities”, many parts of the manuscript are designed logically and systematically. However, there are still some questions for you to answer. And few refinements are needed to be done. The following are the questions and comments of this manuscript:

******Thank you for your invaluable comment. Your understanding of our work is highly appreciated. The manuscript has been improved in clarity and quality as per the reviewer’s comment and suggestion. All queries have been responded and the corretions have been made as highlighted in green color.

Why detect α-amylase inhibitory activity at 25℃, but detect α-glucosidase inhibitory activity and lipase inhibitory activity at 37℃? What is the choose principle?

******Thank you for your comment. The α-amylase inhibitory activity was measured at 37 ºC similar to other enzyme inhibitory activities. Sorry for the typological mistake. The correction has been made in the text. Please see line 225. Authors followed the assays, which were previously used for determining the antidiabetic potential through enzyme inhibitory activities. The reference had been already cited along with the method.

Please provide detailed experimental steps about DPPH-, ABTS-, RSA, FRAP and MCA.

******Thank you for your valuable comment. The detailed experiment steps of all four assays including DPPH-RSA, ABTS-RSA, FRAP, and MCA have been added in the text for a better understanding of readers. Please see line 177-208.

How did the author decide on the different concentrations (10, 5, and 1 mg/mL)?

******Thank you for the insightful comment. The concentrations of samples were based on our preliminary study. In our preliminary study, each sample at 10 mg/mL was completely soluble in all buffer solutions used for antidiabetic activity analyses. As mentioned above, some samples were not completely soluble at the higher concentration. Also, the authors determined the antidiabetic activity of samples at a concentration of 0.5 mg/mL during the preliminary study. No activity was found. Therefore, the authors excluded the 0.5 mg/mL from the main study and selected the sample concentrations of 1, 5, and 10 mg/mL. A reason for choosing the concentrations and other conditions has been given. Please see line 215-219.

I could not figure out the purpose of this study. Why did the author put these two different materials in together? What's the difference? Whether they address the AO, anti-diabetes and AM symptoms through a similar or distinguished approach? The introduction does not explain the choice of raw materials very well.

******Thank you for your insightful comment. In the current study, the authors aimed to enhance the bioactivities of COS through conjugation of polyphenols using ascorbic acid/hydrogen peroxide redox pair reaction with emphasis on food quality improvement and human health benefit. COS is depolymerized product of chitosan obtained from shrimp shell. The effective utilization of COS in the food and pharmaceutical sector could reducd waste disposal-related problems from shrimp processing industries and enhance their market value. Please see line 38-41. The applications of COS were mentioned in the introduction section. Please see line 45-49. Also, the advantages of the consumption of polyphenols were already mentioned in the introduction. Please see line 58-63. Moreover, polyphenols used in this study are commercially available in the market. Therefore, conjugation of those polyphenols can be achived for industry. The related information has been added to the text. Please see line 94-95.

******Authors focused to develop the complete novel functional ingredient, which can serve as an alternative food preservative for shelf-life extension of perishable food and nutraceutical agent for human health benefit. Therefore, authors include antimicrobial (critical for food preservation), antioxidant (important for food preservation and disease caused by reactive oxygen species), and antidiabetic (control type 2 diabetes mellitus) activities in the single study to indicate the versatility of COS-polyphenol conjugates and to prove as the potential food additive and nutraceuticals.

What's the advantage or novelty of this system compared with other chitooligosaccharide conjugates systems?

******Thank you for your comment. The aim of the current study was to enhance the bioactivities of COS through conjugation of different polyphenols using ascorbic acid/hydrogen peroxide redox pair reaction. The authors compared structural characterization and bioactivities including antioxidant, antimicrobial, and antidiabetic between different COS-PPN conjugates prepared in this study. So far there is very limited information or no research available on the chitooligosaccharides conjugate systems has been done intensively. Thus, authors could not compare our study using the existing system with other systems. The novelty statement has been edited for a better understanding of readers. Please see lines 82-92.

Reviewer 3 Report

In the present work, Mittal et al. have analyzed the characteristics, antioxidant, antidiabetic, and antimicrobial activities of chitooligosaccharide conjugates prepared using several phenolic compounds via ascorbic acid/H2O2 free radical grafting.  The work is technical sound and the authors utilized appropriate techniques for the selection of the article. The results are supported by the data and supply useful discussions. There are some typewriting errors and some sentences are rambling. However, following suggestions are recommended:

- In the abstract: “PPNs including gallic, caffeic, and ferulic acids,….” please change with with “phenols and polyphenols including gallic, caffeic, and ferulic acids,…”. These first three compounds aren’t polyphenols. Change also in the other sentences of the manuscript in which there is the same problem.

-In the last paragraph of the introduction, the Author needs to more clearly state the novelty of this paper together with future prospects of this study.

-Authors need to follow the journal format fully in the case of the Reference list. For example, Journal abbreviations, heading, and subheadings etc.

-In the result and discussion section, the author needs to pay more attention and validate their findings with recent previous results and compare if possible.

- The conclusion section must be improved to better explain the obtained results and their potentiality

Author Response

In the present work, Mittal et al. have analyzed the characteristics, antioxidant, antidiabetic, and antimicrobial activities of chitooligosaccharide conjugates prepared using several phenolic compounds via ascorbic acid/H2O2 free radical grafting. The work is technical sound and the authors utilized appropriate techniques for the selection of the article. The results are supported by the data and supply useful discussions. There are some typewriting errors and some sentences are rambling. However, following suggestions are recommended:

******Thank you so much for the invaluable comments and suggestions. All queries have been responded in text. The edited text has been highlighted in blue color.

- In the abstract: “PPNs including gallic, caffeic, and ferulic acids,….” please change with with “phenols and polyphenols including gallic, caffeic, and ferulic acids,…”. These first three compounds aren’t polyphenols. Change also in the other sentences of the manuscript in which there is the same problem.

******Thank you for your insightful review. Polyphenols are secondary metabolites found in plants with one or more hydroxyl groups attached to a phenyl ring, which can be subdivided into 3 main classes, flavonoids, stilbenoids, and phenolic acids. Those classes are further divided into sub-classes, in which flavan-3-ols (sub-class of flavonoids) includes catechin and epigallocatechin gallate while benzoic and hydroxycinnamic acids (sub-class of phenolic acids) comprises gallic acid and caffeic and ferulic acid, respectively. Thus, all phenolic compounds used in this study are broadly considered as polyphenols. Those compounds may be classified into different groups as a function of the number of phenol rings that they contain and of the structural elements that binds these rings to one another. Hence, the authors prefer to use ‘polyphenols’ as an appropriate term to address phenolic compounds that were used in this study. The supporting references have been mentioned below. Please go through it for more details.

Papuc, C., Goran, G. V., Predescu, C. N., Nicorescu, V., & Stefan, G. (2017). Plant polyphenols as antioxidant and antibacterial agents for shelf‐life extension of meat and meat products: Classification, structures, sources, and action mechanisms. Comprehensive Reviews in Food Science and Food Safety, 16(6), 1243-1268.

Cutrim, C. S., & Cortez, M. A. S. (2018). A review on polyphenols: Classification, beneficial effects and their application in dairy products. International Journal of Dairy Technology, 71(3), 564-578.

-In the last paragraph of the introduction, the Author needs to more clearly state the novelty of this paper together with future prospects of this study.

******Thank you for your insightful comment. On the suggestion of the reviewer, the novelty statement has been edited for a better understanding of readers. Also, the prospects of this study have been added in the introduction section for the recognition of work. Please see line 82-92.

-Authors need to follow the journal format fully in the case of the Reference list. For example, Journal abbreviations, heading, and subheadings etc.

******Thank you for your comment. The authors carefully checked the reference list and necessary editing has been done according to the format of the journal. Moreover, all the abbreviations, heading, and subheadings have been checked throughout the text and reference list.

- In the result and discussion section, the author needs to pay more attention and validate their findings with recent previous results and compare if possible.

******Thank you for your insightful comment. In the current study, chitooligosaccharides from shrimp shell chitosan were used at the first time for the conjugation with different polyphenols (gallic acid, catechin, epigallocatechin gallate, ferulic acid, and caffeic acid) using ascorbic acid/hydrogen peroxide redox pair reaction. So far, no literature or study was available on the preparation and characterization of COS conjugate with aforementioned polyphenols. Moreover, there was no report on the antioxidant, antimicrobial, and antidiabetic potential of COS and its polyphenol conjugate. Thus, this study was considered as the novel research. Hence, the authors were not able to compare some results such as conjugation efficiency, total phenol content, antioxidant, and antimicrobial activities with previous findings. However, authors have already validated their findings with previous results in the case of structural characterization and mechanisms involved in the antimicrobial, antioxidant, antidiabetic activities. 

- The conclusion section must be improved to better explain the obtained results and their potentiality

******Thank you for your comment. The conclusion section has been extended with results obtained in this study along with their potential prospects in the future study. Please see line 722-727.
